# Genomic Epidemiology of *Candida auris* in Qatar Reveals Hospital Transmission Dynamics and a South Asian Origin

**DOI:** 10.3390/jof7030240

**Published:** 2021-03-23

**Authors:** Husam Salah, Sathyavathi Sundararaju, Lamya Dalil, Sarah Salameh, Walid Al-Wali, Patrick Tang, Fatma Ben Abid, Clement K. M. Tsui

**Affiliations:** 1Division of Microbiology, Department of Laboratory Medicine and Pathology, Hamad Medical Corporation, Doha, Qatar; HMohamed2@hamad.qa; 2Department of Pathology, Sidra Medicine, Doha, Qatar; ssundararaju@sidra.org (S.S.); ldalil@sidra.org (L.D.); ptang@sidra.org (P.T.); 3Department of Medicine, Division of Infectious Diseases, Hamad Medical Corporation, Doha, Qatar; ssalameh@hamad.qa (S.S.); fabid@hamad.qa (F.B.A.); 4Department of Laboratory Medicine and Pathology, Hamad Medical Corporation, Al Wakra Hospital, Doha, Qatar; walwali@hamad.qa; 5Department of Pathology and Laboratory Medicine, Weill Cornell Medicine-Qatar, Doha, Qatar; 6Division of Infectious Diseases, Faculty of Medicine, University of British Columbia, Vancouver, BC V6T 1Z4, Canada

**Keywords:** candidiasis, candidemia, emerging infectious disease, Middle East, nosocomial outbreak

## Abstract

*Candida auris* is an emerging, multidrug-resistant fungal pathogen that has become a public health threat with an increasing incidence of infections worldwide. *Candida auris* spreads easily among patients within and between hospitals. Infections and outbreaks caused by *C. auris* have been reported in the Middle East region including Oman, Kuwait, Saudi Arabia, and Qatar; however, the origin of these isolates is largely unknown. Pathogen whole genome sequencing (WGS) was used to determine the epidemiology and drug resistance mutations of *C. auris* in Qatar. Forty-four samples isolated from patients in three hospitals and the hospital environment were sequenced by Illumina NextSeq. Core genome single nucleotide polymorphisms (SNPs) revealed that all isolates belonged to the South Asian lineage with genetic heterogeneity that suggests previous acquisition from foreign healthcare. The genetic variability among the outbreak isolates in the two hospitals (A and B) was low. Four environmental isolates clustered with the related clinical isolates, and epidemiologically linked isolates clustered together, suggesting that the ongoing transmission of *C. auris* could be linked to infected/colonized patients and the hospital environment. Prominent mutations Y132F and K143R in *ERG11* linked to increased fluconazole resistance were detected.

## 1. Introduction

Invasive candidiasis is of major public health importance because it is associated with increased mortality, higher healthcare costs, and longer hospital stays compared with other healthcare-associated infections [1]. This problem is compounded by the progressive increase in antifungal resistance among clinically relevant *Candida* spp. such as *C. albicans*, *C. parapsilosis, C. glabrata, C. auris*, and *C. tropicalis* driven by the widespread use of antifungal drugs in human healthcare [1,2].

*Candida auris* has become an emerging opportunistic pathogen, which was first reported in 2009 as an isolate from the external ear of an inpatient at a hospital in Japan [3]. It has since been identified as a cause of nosocomial bloodstream infections (BSI) in numerous countries in East Asia, the Middle East, Africa, the United States, and Europe [4,5,6,7,8,9]. Since *C. auris* is resistant to multiple classes of antifungal agents, able to tolerate temperatures up to 42 °C, and capable of person-to-person transmission and persistence in the hospital environment [10], this easily transmitted and difficult-to-treat yeast has caused outbreaks in many hospitals [11,12].

Whole genome sequencing (WGS) has revealed the presence of five genetic clusters (South Asia, East Asia, Iran, Africa, and South America), which correspond to the geographic distribution [11,13]. The genetic divergence among lineages is large (>1000 single nucleotide polymorphisms (SNPs)), but the intra-lineage variation is small [6]. *Candida auris* has been reported in the Middle East region including Oman, Saudi Arabia, Iran, the UAE, and Kuwait [14,15,16,17,18,19]. Local infections in Qatar have also been reported recently; however, the origin, transmission, and genetic relationships among the isolates were not completely resolved using pulsed-field gel electrophoresis (PFGE) [20]. The underlying genetic mechanisms conferring resistance have also not been characterized. Herein, we use WGS data to determine the possible routes of transmission in Qatar and infer the genetic origins of these isolates in a global context. Mutations related to reduced susceptibility to azoles, amphotericin B, and echinocandins were also characterized from the genomes.

## 2. Materials and Methods

### 2.1. Isolation and Antifungal Susceptibility Assays

*Candida auris* was recovered from specimens collected at three tertiary care hospitals in Doha, Qatar, between 1 April 2018 and 30 November 2020. Clinical specimens were processed according to laboratory standard operative protocol at each microbiology laboratory. *Candida auris* from blood, urine, pleural fluids samples, as well as specimens for *C. auris* screening (axilla, groin, nasal swabs, and environment) were inoculated on chromogenic agar *Candida* (Oxoid, UK) and incubated at 42 °C for five days. Any colonies other than green and blue were identified by MALDI-TOF (Bruker Daltonics, Germany) according to the manufacturer’s protocol of partial extraction.

Antifungal susceptibility patterns were determined on selected clinical and screening isolates using the sensititre YeastOne microdilution method (TREK Diagnostic Systems, Cleveland, OH, USA) following the manufacturer’s instructions.

### 2.2. DNA Extraction and Whole Genome Sequencing

*Candida auris* isolates were streaked on Sabouraud Dextrose Agar (SDA) plates to ensure purity. Genomic DNA was extracted using MasterPure Yeast DNA purification kit (Lucigen Corporation, WI). DNA concentration was measured using Qubit 2 fluorometer (ThermoFisher) and DNA libraries were constructed with a Nextera XT DNA library preparation method (Illumina Inc., San Diego, California, USA) and then sequenced on Illumina NextSeq 550 platform (Illumina Inc.) with 300 cycles (150bp PE) or Illumina Miseq (600 cycles, 300bp PE) at the Integrated Genomics Services Laboratory at Sidra Medicine, Qatar.

### 2.3. Data Analysis

The quality of reads was assessed by Fastqc (https://www.bioinformatics.babraham.ac.uk/projects/fastqc/). The sequence read was trimmed by Trim Galore v0.6.0 (http://www.bioinformatics.babraham.ac.uk/projects/trim_galore/), assembled de novo using SPAdes v.3.9.0, and assessed using QUAST v5.0.2. [21,22]. Contigs of smaller size (<1000 bp) were excluded. Parsnp v1 [23] was used to infer the genetic relationships among the samples using *C. auris* samples from South Asia (assembly accession no. GCA_002759435.2), East Asia (assembly accession no. GCA_003013715.2), Iran (assembly accession no. GCA_016809505.1), South Africa (assembly accession no. GCA_002775015.1), and South America (assembly accession no. GCA_008275145.1). Single nucleotide polymorphism (SNP) analysis was performed using Snippy v4.41 pipeline (https://github.com/tseemann/snippy) using *C. auris* strain B8411 (GenBank accession no. GCA_002759435.2) as the reference genome. Briefly, quality trimmed reads were mapped to the reference genome using BWA v7.17 [24], and the variants were called using samtools v1.9 [25] and Freebayes v1.3 (https://github.com/freebayes/freebayes) with QUAL >30 and DP >10. FastTree (http://meta.microbesonline.org/fasttree/) was used to perform the phylogenetic analysis among samples constructed from the SNPs data. The tree was visualized with Microreact [26]. Mutations in *ERG11, TAC1b, FKS1*, and ORFs [B9J08_00281, B9J08_003025, B9J08_003346] adjacent to *ERG2* linked to azole, echinocandin, and amphotericin B resistance, respectively [27,28,29] were retrieved from genome assemblies and examined through comparative genomics and sequence analysis using MEGA X [30].

## 3. Results

### 3.1. Genomic Epidemiology

We sequenced a total of 44 *Candida auris* genomes, of which 40 were from humans and four were from the hospital environment. These isolates were from 36 patients, 13 from Hospital A, 22 from Hospital B, and 1 from Hospital C (Table 1). Three isolates were from invasive infections, while thirty-seven isolates were colonizers. Sequencing using the Illumina system generated 5139210 to 39112452 high-quality reads in each sample: sequencing depth ranged from 54 to 262 (median = 89) (Appendix A).

Whole genome sequence data using core genome SNPs indicated that all isolates belonged to the South Asian lineage (Appendix A). The high-resolution SNP tree revealed a small level of genetic heterogeneity among the isolates (Figure 1). Thirty-seven isolates from two tertiary care hospitals (A and B) including both human and environmental samples in various medical wards and ICUs collected from 2018 to 2020 clustered in one major clade, which could be the predominant “circulating clone”, while three isolates (CAS29, CAS20044, and CAS3357) from one patient in Hospital C clustered separately. Four other isolates (CAS20, CAS12, CAS16, CAS17) from Hospitals A and B also diverged from the major clade. CAS20 represented the first reported case in Qatar: the patient had received healthcare in Oman, and the sequence was divergent from the outbreak samples in Hospitals A and B. The remaining three divergent isolates (CAS12, 16, and 17) were from patients without recent travel histories, while CAS12 was from a patient who was transferred from a long-term care unit within the state of Qatar, indicating that the hospital/community may harbor additional circulating *C. auris* strain from the South Asian lineage, apart from the predominant outbreak clone.

Within the major clade containing 33 patients from both outbreaks and sporadic samples, the genetic difference was very small (range from 0 to 19 bp, median = 5bp); the results of only five pairwise comparisons were above 15 bp. The variations among the isolates from the same patient on different occasions in Hospital C were also small (<5 bp). Four environmental samples were 100% identical to the respective clinical isolates (Figure 1). The high degree of sequence similarity among isolates within and between Hospitals A and B supports the report of a widespread *C. auris* outbreak in Qatar. Although the SNP tree suggests the presence of subclades within the major clade, these clades do not have strong statistical support, and the SNP variation was small. Of the patients reviewed, three had visited or come from a foreign country (Oman, Syria, Sudan, or India) (Table 1). Six patients had been transferred to and from different hospitals and home care units (Table 1, Figure 1).

### 3.2. Antifungal Susceptibility

Currently, there are no established susceptibility breakpoints for *C. auris*. Antifungal susceptibilities were determined for 11 isolates (selected based on clinical significance and the hospital), and their minimum inhibitory concentrations (MIC) were interpreted according to the Center for Disease Control (CDC) tentative breakpoints (https://www.cdc.gov/fungal/candida-auris/c-auris-antifungal.html (accessed on 26 January 2021)). The MICs to fluconazole were above the breakpoints for all isolates (≥32 mg/L) according to the guidelines, and three isolates had elevated MICs to more than one azole drug (Table 2). Ten isolates had elevated MICs to amphotericin B (MIC = 2 mg/L), while one isolate (CAS12) exhibited a high MIC to caspofungin (MIC = 8 mg/L) (Table 2).

Mutations in *ERG11* and *TAC1B* that were associated with reduced susceptibility to fluconazole were examined from all the genome assemblies; 42 isolates carried Y132F mutations, while two isolates (CAS17 and CAS20) had substitution K143R (Table 2). CAS17 and CAS20 also possessed mutation A640V in *TAC1B*. All isolates had substitutions R252T and I44F in B9J08_003025 [*ERG24*] and B9J08_003346 [*ERG29*], respectively, which may be associated with amphotericin B resistance [28], while mutation G145D in B9J08_00281 [*ERG28*] was not identified. *FKS1* S639F mutation associated with echinocandin resistance was not detected.

## 4. Discussion

All *Candida auris* isolates in Qatar belong to the South Asian lineage, which is similar to isolates identified in Saudi Arabia and Oman [14,16]. The largest proportion of the expatriate populations in Qatar and other Gulf Cooperation Council countries is from the Indian subcontinent. The fungal pathogen may have been introduced through carriage within the expatriate workforce or residents who are seeking medical care in the Indian subcontinent. However, there were no isolates belonging to the African lineage despite many workers being from Africa. For the isolates that were genetically different from the other outbreak-related isolates, one patient received hospital care in Oman, and another patient had healthcare exposure in Sudan and India. The results also underscore the potential role of international travel between countries as a mode of *C. auris* dissemination and colonization [6,31].

Our study confirmed the high level of clonality among the *C. auris* outbreak isolates [6,7], as the pairwise SNP differences was small (1–19 bp, median = 5) among 37 isolates from two major tertiary hospitals. Additionally, 12 of 13 isolates from Hospital A collected in 2019 clustered with three isolates from Hospital B collected in 2019 and 2020. The clustering of these isolates between the two hospitals suggests that patients transferred between facilities could also be a source of transmission. Chow et al. proposed a genetic distance of < 12 between patients as an indication of recent transmission [6]. The presence of several subclades within the largest clade may indicate continued pathogen microevolution during the ongoing local outbreak. Intra-host variation such as (CAS25 and CAS32) may also indicate microevolution.

Environmental screening as part of the outbreak analysis confirmed the presence of *C. auris* in the hospital environment, and the genetic relatedness of clinical and environmental samples indicates possible cross-transmission between patients and the hospital environment by patients and *vice versa*. *C. auris* is well known for its capability to survive and persist in the healthcare settings [32] and cause ongoing transmission within the hospital environment [4,17,33]. Once *C. auris* is disseminated in the hospital environment, it is difficult to eradicate, and hospital contamination has been reported in many countries including the USA, India, Colombia, and the UK [4,34,35,36]. Most patients did not travel outside Qatar in the past 6 months before infection, suggesting that the healthcare facilities could be the reservoir of this pathogen, and the cases were acquired from within the country. Our findings are consistent with other reports that patients who are hospitalized in the ICU, critically ill, or have undergone surgery are at greater risk of colonization and infections [12].

Despite a high degree of clonality, we identified genetic polymorphism among the isolates within the South Asian lineage. Four clinical isolates were more divergent than the 37 samples clustered in the major clade, implying other unsampled chains of transmission; *C. auris* not belonging to the predominant circulating clone may be carried/colonized by asymptomatic patients and resident in the community. Whole genome comparison indicated that these samples were more similar to other *C. auris* samples from India, Pakistan, and Saudi Arabia (personal communications). Intra-lineage variation in the South Indian lineage has also been demonstrated in Saudi Arabia and Oman using short tandem repeat (STR) typing techniques and WGS [6,14,37]. This variation could also reflect the ongoing microevolution and adaptation in the clinical and natural environment [38].

*Candida auris* is a significant challenge to healthcare in Qatar and elsewhere because it is multidrug resistant. All the local isolates were resistant to fluconazole, which was largely due to either T312F or K143R mutations in *ERG11*, and A640V in *TAC1b*, which are common in the South Asian isolates [27,39]. *ERG11* encodes the cytochrome P450 lanosterol 14α-demethylase targeted by the azoles, while *TAC1b* encodes a transcription factor that can regulate the expression of *CDR1,* an ATP-binding cassette (ABC)-type efflux pump-encoding gene [27]. Three isolates were resistant to other azoles, which was possibly due to the increased copy number of *ERG11*, gene duplication, and transporter gene family expansion [38]. Most *C. auris* isolates also had reduced susceptibility to amphotericin B, despite the absence of substitution G145D in B9J08_00281 [*ERG28*] [28]. The differential expression of *ERG* families and mutations in certain genes may contribute to the increased resistance [28,38]; however, the mechanism of amphotericin B resistance in *C. auris* is not completely understood. Although no mutation was detected in *FKS1*, which encodes the subunits of 1,3-beta-D-glucan synthase in the fungal cell wall targeted by the echinocandins [29,39], one isolate was resistant to caspofungin. Several isolates were resistant to at least two classes of antifungals [20], thus limiting treatment options. One of the limitations in this study was that an antifungal susceptibility test was not performed by a reference method (i.e., Clinical & Laboratory Standards Institute (CLSI) or European Committee on Antimicrobial Susceptibility Testing (EUCAST)). Additional work will be required to investigate the mechanisms of drug resistance, to identify the candidate mutations and genes, and to determine the virulence of these *C. auris* isolates. More investigations are ongoing to determine the optimal management and control of *C. auris* infections in Qatar.

## Figures and Tables

**Figure 1 jof-07-00240-f001:**
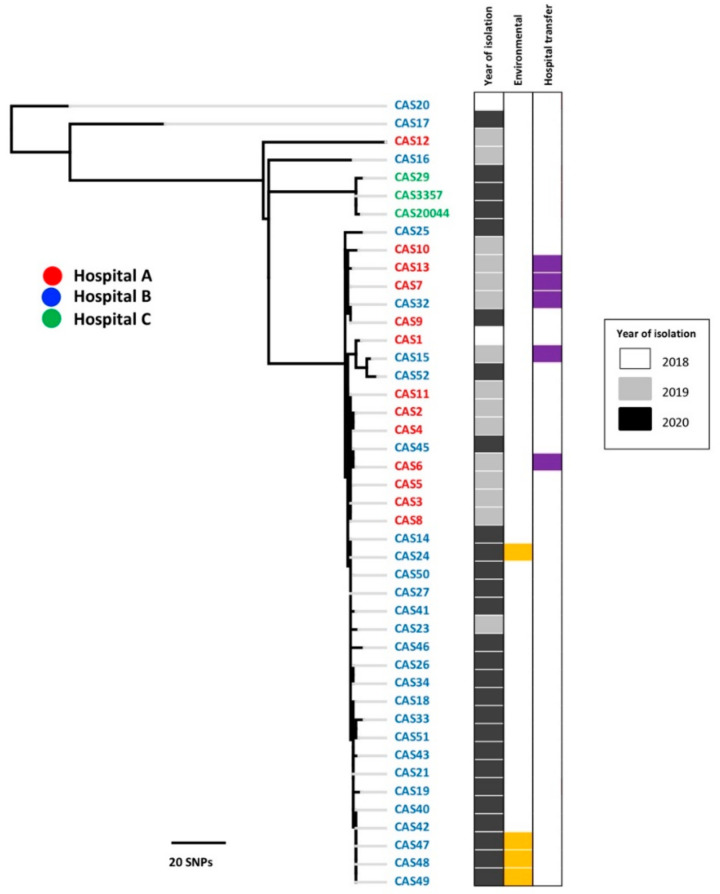
Genetic relationships of the 44 *Candida auris* isolates. Comparison of all Qatar isolates based on high-resolution variants. The color blocks highlight the year of isolation, environmental isolates, and hospital transfer.

**Table 1 jof-07-00240-t001:** Clinical epidemiology of the *Candida auris* isolates.

Patient Number	Isolate Code	Date of Isolation	Specimen	Hospital	Unit	Nationality	Transfer between Hospitals in Qatar	Admission to Hospital Abroad (within 6 Months)
1	CAS20	12-May-18	Urine	B	Medical ward	Omani	no	Yes (Oman)
2	CAS1	21-Dec-18	ETT ^a^	A	ICU ^b^	Pakistani	no	no
3	CAS2	27-Feb-19	Wound swab	A	ICU	Qatari	no	no
4	CAS3	17-Mar-19	Wound swab	A	ICU	Qatari	no	no
5	CAS4	26-Jun-19	Nasal swab	A	ICU	Indian	no	no
6	CAS5	09-Jun-19	Nasal swab	A	High dependency unit	Palestinian	no	no
7	CAS6	18-Jun-19	Nasal swab	A	ICU	Filipino	B/A	no
8	CAS7	18-Jun-19	Screening swab ^c^	A	Medical ward	Qatari	A/B	no
9	CAS8	28-Jun-19	Skin swab	A	Medical ward	Qatari	no	no
10	CAS9	28-Jun-19	Nasal swab	A	Medical ward	Qatari	no	no
11	CAS10	28-Jun-19	Nasal swab	A	Medical ward	Pakistani	no	no
12	CAS11	02-Jul-19	Skin swab	A	LTCU ^d^	Nepalese	A/LTCU *	no
13	CAS12	11-Jul-19	Screening swab	A	Medical ward	Indian	no	no
14	CAS13	30-Jul-19	Groin swab	A	Medical ward	Qatari	A/B	no
15	CAS16	04-Aug-19	Urine	B	Cardiac ICU	Palestinian	no	no
16	CAS23	01-Sep-19	Groin swab	B	Medical ward	Qatari	no	no
17	CAS32	09-Sep-19	Urine	B	Surgical ward	Qatari	B/A	no
17	CAS15	14-Sep-19	Urine	B	Surgical ward	Qatari	B/A	no
18	CAS14	04-Jan-20	ETT	B	Medical ward	Qatari	no	no
(18)	CAS24	15-Jan-20	Bedside table (patient 18)	B	ICU (203-1)	N/A	N/A	N/A
19	CAS18	08-Jan-20	Nasal swab	B	ICU	Qatari	no	no
20	CAS21	08-Jan-20	Groin swab	B	Medical ward	Qatari	no	no
21	CAS19	09-Jan-20	Axilla swab	B	ICU	Syrian	no	Yes (Syria)
22	CAS26	20-Feb-20	BAL ^e^	B	ICU	Nepalese	no	no
22	CAS33	20-Feb-20	Pleural fluid	B	ICU	Nepalese	no	no
23	CAS25	24-Feb-20	Pus	B	LTCU	Qatari	no	no
24	CAS27	17-Mar-20	Blood	B	Medical ward	Indian	no	no
25	CAS34	17-Jun-20	Blood	B	ICU	Bangladeshi	no	no
26	CAS40	08-Jul-20	Axilla swab	B	Medical ward	Qatari	no	no
27	CAS50	20-Jul-20	Nasal swab	B	LTCU	Indian	no	no
28	CAS41	18-Aug-20	Urine	B	Medical ward	Palestinian	no	no
29	CAS52	25-Aug-20	Nasal swab	B	LTCU	Qatari	no	no
30	CAS17	28-Aug-20	Axilla swab	B	Medical ward	Qatari	no	no
31	CAS42	14-Sep-20	Groin swab	B	Medical ward	Omani	no	no
(31)	CAS47	16-Sep-20	Bed (patient 31)	B	Medical ward	N/A	N/A	N/A
(31)	CAS48	16-Sep-20	Couch (patient 31)	B	Medical ward	N/A	N/A	N/A
(31)	CAS49	16-Sep-20	Cabinet (patient 31)	B	Medical ward	N/A	N/A	N/A
32	CAS43	14-Sep-20	Axilla swab	B	Medical ward	Iranian	no	no
33	CAS51	14-Sep-20	Axilla swab	B	LTCU	Qatari	no	no
34	CAS45	14-Oct-20	Groin swab	B	LTCU	Qatari	no	no
35	CAS46	02-Nov-20	Axilla swab	B	Medical ward	Indian	Home care/B	no
36	CAS20044	12-May-20	BAL	C	Oncology ward	Sudanese	no	Yes (Sudan, India)
36	CAS3357	20-May-20	Screening swab	C	Oncology ward	Sudanese	no	Yes (Sudan, India)
36	CAS29	17-Jun-20	Tracheal aspirate	C	Oncology ward	Sudanese	no	Yes (Sudan, India)

^a^ Endotracheal tube aspirate, ^b^ Intensive care unit, ^c^ Axilla, groin, or nasal swab, ^d^ Long-term care unit, **^e^** Bronchoalveolar lavage, * LTCU: Long term care unit.

**Table 2 jof-07-00240-t002:** In vitro susceptibility to nine antifungal agents and *ERG11* mutations in selected *C. auris* isolates. Elevated minimum inhibitory concentration (MIC) values are bold. AMB = amphotericin B; 5FC = flucytosine, CAS = caspofungin, FLC = fluconazole, ITC = itraconazole, VOR = voriconazole, POS = posaconazole, ANI = anidulafungin, and MICA = micafungin.

Isolate	FLC	ITC	POS	VOR	AMB	5FC	CAS	ANI	MICA	*ERG11* Mutations
CAS12	**256**	0.25	0.12	1	1	0.12	**8**	0.5	0.5	Y132F
CAS14	**32**	0.06	0.03	0.12	**2**	0.06	0.25	0.25	0.12	Y132F
CAS16	**128**	0.5	0.25	**8**	**2**	0.06	0.5	0.5	0.5	Y132F
CAS17	**128**	0.25	0.12	1	**2**	0.06	0.12	0.25	0.12	K143R
CAS20	**256**	0.5	0.25	2	**4**	0.12	0.25	0.12	0.12	K143R
CAS25	**128**	**16**	**8**	**8**	**2**	0.12	0.5	0.25	0.12	Y132F
CAS27	**64**	0.25	1	1	**2**	0.12	0.5	0.25	0.12	Y132F
CAS33	**128**	0.12	0.12	**8**	**2**	0.06	0.5	0.25	0.25	Y132F
CAS34	**128**	0.12	0.03	0.25	**2**	0.06	0.25	0.12	0.12	Y132F
CAS41	**32**	0.06	0.015	0.12	**2**	0.06	0.06	0.12	0.06	Y132F
CAS20044	**128**	0.12	0.06	0.5	**2**	0.12	0.25	0.12	0.12	Y132F

## Data Availability

The raw sequencing reads are available from the National Center for Biotechnology Information (NCBI) under the accession number PRJNA693430.

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
