# Peer review of "Genomic Epidemiology of *Candida auris* in Qatar Reveals Hospital Transmission Dynamics and a South Asian Origin"

_jof, 2021, doi:10.3390/jof7030240_

Round 1

Reviewer 1 Report

This manuscript describes the epidemiology of C. auris in Qatar. The subject is timely and suitably addressed. Despite minor corrections, the paper is well written. The most worrying aspect is that the results are rather confirmatory of what was previously known, e.g. no new mutations were found, no new mechanisms of antifungal resistance were explored. 

Despite the isolates have known ERG11 mutations, the vast majority is only resistant to fluconazole and not to the other azoles tested. Since the cellular target is the same for all azole class of antifungals, I would like the authors to comment about that and what they think to be one possible explanation for this to happen.

If all isolates had substitutions R252T 17 and I44F in B9J08_003025 and B9J08_003346 respectively, which may be associated with amphotericin B resistance, why the isolate CAS12 does not present resistance to Amphotericin B (table 2)?

Minor corrections:

Line 27: which hospitals? You refer three hospitals above (line 24).

Line 47: °C

Line 67: Candida auris should be written in italic.

Line 68: Candida should be written in italic.

Line 68: °C

Line 70: Antifungal, not “anti-fungal”. This part should be separated from “Isolation” section, since it has nothing to do with the isolation process. Or you may change the section title to include antifungal susceptibility assays.

Line 86: de novo should be written in italic.

Line 98: parenthesis in the ref 26 should be removed.

Line 99: please indicate the names of the adjacent ORFs that you are referring. Further on, when you refer those ORFs (line 18 after figure 1) it is not clear that you are referring to these ORFs adjacent to ERG2. Also, you should indicate which genes are related to which antifungal resistance, e.g.: Mutations in ERG11, TAC1b, FKS1 and ORFs adjacent to ERG2 linked to azole, echinocandin and Amphotericin B resistance, respectively (…)

Line 116: indicate which hospitals.

Figure S1:

  • Figure must have a legend.
  • auris must be written in italic.
  • Why is the “Iran” branch so apart? Maybe you should provide some insights into the genetic divergence among the branches.
  • Samples “from Qatar” and not “In Qatar”.

Table 1:

  • Patient 8 and 36 have two samples (Axilla/groin swab)? So, two isolates?

 (after Figure. 1)

Line 6: why were those isolates chosen? What was the criteria? Was it random?

Line 7: MIC what? 50? 

Line 7: high based on what ground?

Line 9: MIC, singular.

Line 27: Do you mean “Despite many workers being from Africa, there were no isolates from African origin”?

Line 61: “indicated that (…)”

Line 63: write down STR: Short Tandem Repeat (STR) typing.

References

  • Besides references 31, 34, 36, 37, 38 and 39, all the references have Candida or Candida auris written normally. It must be correct to italic.
  • Some references (2, 10, 14, 17, 27, 28 and 29) have de DOI, the remain have not. You should delete the DOI in these references or include the DOI of the remain.
  • Reference 12: year must be bold as in the remain references.
  • References 27, 29 and 39: genes’ name should be written in italic.
  • Reference 31: it has written “Author correction” (??)
  • Reference 33: volume? Issue? Pages?

Author Response

Reviewer#1

This manuscript describes the epidemiology of C. auris in Qatar. The subject is timely and suitably addressed. Despite minor corrections, the paper is well written. The most worrying aspect is that the results are rather confirmatory of what was previously known, e.g. no new mutations were found, no new mechanisms of antifungal resistance were explored. 

Thank you for the suggestions and questions. The aim of the investigation was to study the transmission and spread of C. auris in Qatar by WGS data. We will perform further investigation on the molecular mechanisms leading to antifungal resistance when all AST data become available.

Despite the isolates have known ERG11 mutations, the vast majority is only resistant to fluconazole and not to the other azoles tested. Since the cellular target is the same for all azole class of antifungals, I would like the authors to comment about that and what they think to be one possible explanation for this to happen.

We have added a sentence “Three isolates were resistant to other azoles, possibly due to the increased copy number of ERG11, gene duplication and transporter gene family expansion [38].” In the discussion.

If all isolates had substitutions R252T 17 and I44F in B9J08_003025 and B9J08_003346 respectively, which may be associated with amphotericin B resistance, why the isolate CAS12 does not present resistance to Amphotericin B (table 2)?

We have included a sentence “Differential expression of ERG families and mutations in certain genes may contribute to the increased resistance [28,38]; however, the mechanism of amphotericin B resistance in C. auris is not completely understood.” In the discission.

Minor corrections:

Line 27: which hospitals? You refer three hospitals above (line 24).

Added.

Line 47: °C

Revised

Line 67: Candida auris should be written in italic.

Revised

Line 68: Candida should be written in italic.

Revised

Line 68: °C

Revised

Line 70: Antifungal, not “anti-fungal”. This part should be separated from “Isolation” section, since it has nothing to do with the isolation process. Or you may change the section title to include antifungal susceptibility assays.

Revised. We also revised the section title.

Line 86: de novo should be written in italic.

Revised

Line 98: parenthesis in the ref 26 should be removed.

Revised

Line 99: please indicate the names of the adjacent ORFs that you are referring. Further on, when you refer those ORFs (line 18 after figure 1) it is not clear that you are referring to these ORFs adjacent to ERG2. Also, you should indicate which genes are related to which antifungal resistance, e.g.: Mutations in ERG11, TAC1b, FKS1 and ORFs adjacent to ERG2 linked to azole, echinocandin and Amphotericin B resistance, respectively (…)

Revised.

Line 116: indicate which hospitals.

Revised.

Figure S1:

  • Figure must have a legend.

The legend was written in the supplementary section.

  • aurismust be written in italic.

Revised

  • Why is the “Iran” branch so apart? Maybe you should provide some insights into the genetic divergence among the branches.

In the original article, the Iran sample was mentioned to be divergent. So we would not discuss its divergence in this manuscript.

  • Samples “from Qatar” and not “In Qatar”.

Revised.

Table 1:

  • Patient 8 and 36 have two samples (Axilla/groin swab)? So, two isolates?

Sorry for the confusion. Only one sample in these two patients. We revised the table to avoid confusion.

(after Figure. 1)

Line 6: why were those isolates chosen? What was the criteria? Was it random?

We added “selected based on clinical significance and the hospital)” there.

Line 7: MIC what? 50? 

Yes.

Line 7: high based on what ground?

The sentenced was revised to avoid confusion.

Line 9: MIC, singular.

Revised

Line 27: Do you mean “Despite many workers being from Africa, there were no isolates from African origin”?

Revised. We wanted to say no isolate was from the African lineage.

Line 61: “indicated that (…)”

Revised

Line 63: write down STR: Short Tandem Repeat (STR) typing.

Revised

References

  • Besides references 31, 34, 36, 37, 38 and 39, all the references have Candidaor Candida auris written normally. It must be correct to italic.
  • Some references (2, 10, 14, 17, 27, 28 and 29) have de DOI, the remain have not. You should delete the DOI in these references or include the DOI of the remain.
  • Reference 12: year must be bold as in the remain references.
  • References 27, 29 and 39: genes’ name should be written in italic.
  • Reference 31: it has written “Author correction” (??)
  • Reference 33: volume? Issue? Pages?

All the suggestions to the reference list have been revised and corrected.

Reviewer 2 Report

General comments

In this paper, the authors have evaluated the molecular epidemiology of Candida auris in Qatar. The aim of the study is of interest as C. auris is an emerging drug-resistant pathogen that causes hospital outbreaks. The epidemiology is poorly known and any advances in this area are welcome.

WGS has been used to study 40 clinical as well as 4 environmental isolates retrieved at three hospitals between 2018 and 2020.

Overall, the results show that all isolates belong to the South Asian clade and harbored molecular markers of antifungal resistance.

In summary, this is an important study that provide useful data.

Specific comments

  1. Lines 89-91: Please state that the numbers are GenBank assembly accession.
  2. Line 90: Is IFC2087 an accession number or an isolate identification number? Please clarify.
  3. Line 104: Please state if there were C. auris invasive infections or if all isolates were colonizers.
  4. Lines 132-133: It is stated that 2 patients had visited or come from a foreign country. Please state which countries.
  5. Table 1: Nationality is perhaps not the best parameter. It would be more informative to know if the patients had stay abroad in the 3 to 6 months before hospitalization.
  6. Antifungal susceptibility: AFST was not performed by a reference method (i.e. CLSI or EUCAST). This point must be included in the Discussion section as a limitation of the study.
  7. Page 7, line 18: Please state what is the putative gene and function of these sequences if known.

Author Response

Reviewer #2

General comments

In this paper, the authors have evaluated the molecular epidemiology of Candida auris in Qatar. The aim of the study is of interest as C. auris is an emerging drug-resistant pathogen that causes hospital outbreaks. The epidemiology is poorly known and any advances in this area are welcome.

WGS has been used to study 40 clinical as well as 4 environmental isolates retrieved at three hospitals between 2018 and 2020.

Overall, the results show that all isolates belong to the South Asian clade and harbored molecular markers of antifungal resistance.

In summary, this is an important study that provide useful data.

Thank you for the encouraging response.

Specific comments

  1. Lines 89-91: Please state that the numbers are GenBank assembly accession. ADDED to the results
  2. Line 90: Is IFC2087 an accession number or an isolate identification number? Please clarify. Replaced by the accession number
  3. Line 104: Please state if there were C. auris invasive infections or if all isolates were colonizers. A sentence was added to the result session.
  4. Lines 132-133: It is stated that 2 patients had visited or come from a foreign country. Please state which countries. The countries were listed in the table, but now included in the result session as well.
  5. Table 1: Nationality is perhaps not the best parameter. It would be more informative to know if the patients had stay abroad in the 3 to 6 months before hospitalization. The patients had stayed within 6 months (added to the last column in the table)
  6. Antifungal susceptibility: AFST was not performed by a reference method (i.e. CLSI or EUCAST). This point must be included in the Discussion section as a limitation of the study. The sentence has been added to the discussion.
  7. Page 7, line 18: Please state what is the putative gene and function of these sequences if known. The functions of the genes have been added in the discussion as part of responses to the questions raised by reviewer #1. Therefore the putative gene function was not repeated in the result session.

Reviewer 3 Report

Minor comments:

  • Please revise Candida auris always appears in italics through the manuscript. For example, it doesn’t at line 67.
  • Looks like there’s some editing error where the ° at 42 °C is missing. Please check

Author Response

Reviewer #3

Minor comments:

  • Please revise Candida auris always appears in italics through the manuscript. For example, it doesn’t at line 67.

Revised

  • Looks like there’s some editing error where the ° at 42 °C is missing. Please check

Revised

Round 2

Reviewer 1 Report

Thank you for the complied corrections. It has been significantly improved.  

In the discussion, the pharagraph where you describe the function of the genes related to drug resistance is ok. Despite not being the main focus of the work, it was actually missing in the previous version since you decided to address drug susceptibility. 

Best wishes